# Photophysical and Electrocatalytic Properties of Rhenium(I) Triazole-Based Complexes

**Adrian Comia [1], Luke Charalambou [1], Salem A. E. Omar [1], Paul A. Scattergood [1,2,*], Paul I. P. Elliott [1,2,*] and Alessandro Sinopoli [3,*]**

[1] Department of Chemistry, University of Huddersfield, Queensgate, Huddersfield HD1 3DH, UK; acomia07@gmail.com (A.C.); luke.c96@LIVE.CO.UK (L.C.); s.omar@hud.ac.uk (S.A.E.O.)

[2] Centre for Functional Materials, University of Huddersfield, Queensgate, Huddersfield HD1 3DH, UK

[3] Qatar Environmental & Energy Research Institute, Hamad Bin Khalifa University, Doha 34110, Qatar

\* Correspondence: p.scattergood@hud.ac.uk (P.A.S.); p.i.elliott@hud.ac.uk (P.I.P.E.); asinopoli@hbku.edu.qa (A.S.); Tel.: +44-1484 -472174 (P.A.S.); +44-1484-472320 (P.I.P.E.); +974-4454-0157 (A.S.)

**Abstract:** A series of $[Re(N^\wedge N)(CO)_3(Cl)]$ ($N^\wedge N$ = diimine) complexes based on 4-(pyrid-2-yl)-1,2,3-triazole (**1**), 1-benzyl-4-(pyrimidin-2-yl)-1,2,3-triazole (**2**), and 1-benzyl-4-(pyrazin-2-yl)-1,2,3-triazole (**3**) diimine ligands were prepared and their photophysical and electrochemical properties were characterized. The ligand-based reduction wave is shown to be highly sensitive to the nature of the triazole-based ligand, with the peak potential shifting by up to 600 mV toward more positive potential from **1** to **3**. All three complexes are phosphorescent in solution at room temperature with $\lambda_{max}$ ranging from 540 nm (**1**) to 638 nm (**3**). Interestingly, the complexes appear to show inverted energy-gap law behaviour ($\tau$ = 43 ns for **1** versus 92 ns for **3**), which is tentatively interpreted as reduced thermal accessibility of metal-centred ($^3$MC) states from photoexcited metal to ligand charge transfer ($^3$MLCT) states upon stabilisation of the $N^\wedge N$-centred lowest unoccupied molecular orbital (LUMO). The photophysical characterisation, supported by computational data, demonstrated a progressive stabilization of the LUMO from complex **1** to **3**, which results in a narrowing of the HOMO–LUMO energy gap (HOMO = highest occupied molecular orbital) across the series and, correspondingly, red-shifted electronic absorption and photoluminescence spectra. The two complexes bearing pyridyl (**1**) and pyrimidyl (**2**) moieties, respectively, showed a modest ability to catalyse the electroreduction of $CO_2$, with a peak potential at ca. $-2.3$ V versus Fc/Fc$^+$. The catalytic wave that is observed in the cyclic voltammograms is slightly enhanced by the addition of water as a proton source.

**Keywords:** rhenium; complexes; triazole; ligands; luminescence; electrocatalysis

## 1. Introduction

The demand for energy by an ever-growing population has seen atmospheric levels of $CO_2$ rise to levels unprecedented in human history with the effects of anthropogenic climate change already evident. As part of an attempt to combat this, significant effort has been put into the development of catalytic systems for the photochemical and/or electrochemical conversion of $CO_2$ into useful chemical feedstocks. Transition metal carbonyl complexes, such as $[Re(bpy)(CO)_3Cl]$, have attracted attention for their ability to photocatalytically and electrocatalytically reduce $CO_2$ to carbon monoxide and other $C_1$ compounds [1–4].

The development of novel catalytic systems for these applications has involved significant efforts in ligand design in attempts to improve efficiency. Therefore, exploring facile routes to ligand synthesis that are robust and versatile could enable access to a wide expanse of chemical designs. In this regard, the copper(I) catalysed 1,3-dipolar cycloaddition of alkynes and azides to form 1,4-disubstituted-1,2,3-triazoles has become a popular transformation in the design of ligands for transition metal complexes over the past decade [5–8]. This has resulted in several reports on the preparation of 1,2,3-triazole-based rhenium(I) complexes [9–20], which have been applied for their therapeutical activity [21–25] and as luminescent biological imaging microscopy probes [26–34]. Recently, Ching et al. reported a series of complexes of the form [Re(N^N)(CO)$_3$Cl], where the N^N ligand is based on a 4-(pyrid-2-yl)-1,2,3-triazole framework [35]. These complexes proved to be efficient electrocatalysts for $CO_2$ reduction with good stability, in particular the complex with a 1-(2,4,6-tri-*tert*-butylphenyl) substituent on the triazole ring.

On the basis of our interest in triazole-based coordination chemistry, we report the synthesis, characterisation, and photophysical properties of the new rhenium(I) complexes [Re(pymtz)(CO)$_3$Cl] (**2**; pymtz = 1-benzyl-4-(pyrimidin-2-yl)-1,2,3-triazole) and [Re(pyztz)(CO)$_3$Cl] (**3**; pyztz = 1-benzyl-4-(pyrazin-2-yl)-1,2,3-triazole) (Figure 1), as well as results from their application in the electrocatalytic reduction of $CO_2$. The replacement of the pyridine donor in the pytz ligand of **1** with the pyrimidine and pyrazine donors in **2** and **3**, respectively, progressively stabilised the ligand-centred LUMO (moving closer in energy to that of the archetypal model complex and known electrocatalyst [Re(bpy)(CO)$_3$Cl]), and thus **2** and **3** will be easier to reduce. We reasoned that this increased ease of electrochemical reduction with respect to complexes based on the structure of **1** might then make these new complexes more potent electrocatalysts, but would also enable the exploration of the photophysical properties of these new materials.

**Figure 1.** Structures of rhenium(I) complexes **1**, **2**, and **3**.

## 2. Results

### *2.1. Ligand Synthesis and Characterization*

The ligands 1-benzyl-4-(pyrimidin-2-yl)-1,2,3-triazole (pymtz) and 1-benzyl-4-(pyrazine-2-yl)-1,2,3-triazole (pyztz) were prepared as previously reported [36]. The complexes [Re(N^N)(CO)$_3$Cl] (N^N = pymtz (**2**); pyztz (**3**)) were prepared through heating to reflux the ligands in the presence of [Re(CO)$_5$Cl] in toluene and were isolated in good to moderate yields. Complexes **2** and **3** were characterised by [1]H and [13]C NMR spectroscopy (Figures S1–S4) and electrospray mass spectrometry. The [1]H NMR spectra of complexes **2** and **3** show three unique environments for the pyrimidinyl/pyrazinyl fragment, a distinctive triazole singlet resonance, and methylene and aromatic ring proton resonances of the benzyl fragment (Supplementary Materials).

## 2.2. Cyclic Voltammetry

Cyclic voltammetry experiments were conducted on complexes **1**–**3** to measure the electronic effect of the different *N*-heterocyclic ligands. The redox potentials are tabulated in Table 1. Cyclic voltammograms for each complex are shown in Figure 2. The electrochemical behaviour of the various complexes, at a concentration of 1 mM, was investigated by cyclic voltammetry (CV) at 0.1 V·s$^{-1}$ in nitrogen-saturated acetonitrile in the presence of Bu$_4$NPF$_6$ (0.2 M) as the supporting electrolyte, using a glassy carbon electrode. Scanning toward positive potentials, a first irreversible oxidation is observed at ~1.0 V versus Fc/Fc$^+$, associated with the Re(II)/Re(I) couple. This feature is only very slightly sensitive to modification of the ligands; as expected, slightly more positive values are obtained in the case of **2** and **3**, because of the increased electron withdrawing ability of the coordinated six-membered *N*-heterocyclic fragment over that of pyridine. Scanning toward negative potentials, an irreversible feature is observed at −2.12 V for **1**, −1.89 V for **2**, and −1.56 V for **3** versus Fc/Fc$^+$, all of which are assigned to reduction of the NˆN donor fragment. The redox potentials reported herein are in line with those for similar rhenium–triazole complexes investigated by Ching et al. and with data from our laboratory for osmium(II) and iridium(III) complexes bearing pyridyl- versus pyrazinyltriazole ligands [36,37]. In agreement with the stabilisation of the pyrimidine- and pyrazine-localised LUMOs for **2** and **3**, respectively, relative to that of **1**, the electrochemical reduction features of **2** and **3** are anodically shifted with respect to the corresponding pyridyltriazole complex (**1**). Furthermore, CVs of **3** display an additional feature when scanning to further cathodic potential at ca. −2.02 V versus Fc/Fc$^+$, which was possibly assigned to the Re(I)/Re(0) wave [38,39].

**Table 1.** Summarised electrochemical data for complexes **1** to **3** (1 mM of compounds **1**, **2**, and **3** in MeCN with 0.2 M Bu$_4$NPF$_6$ as the supporting electrolyte, recorded under N$_2$ at room temperature (r.t.) at 0.1 V·s$^{-1}$ with a glassy carbon disk electrode).

| Complex | E$_{ox}$/V | E$_{red}$/V | $(i_c/i_p)^2$ |
|---|---|---|---|
| **1** | +0.99 | −2.12 | 13 |
| **2** | +1.05 | −1.89 | 8 |
| **3** | +1.06 | −1.56, −2.02 | 5 |

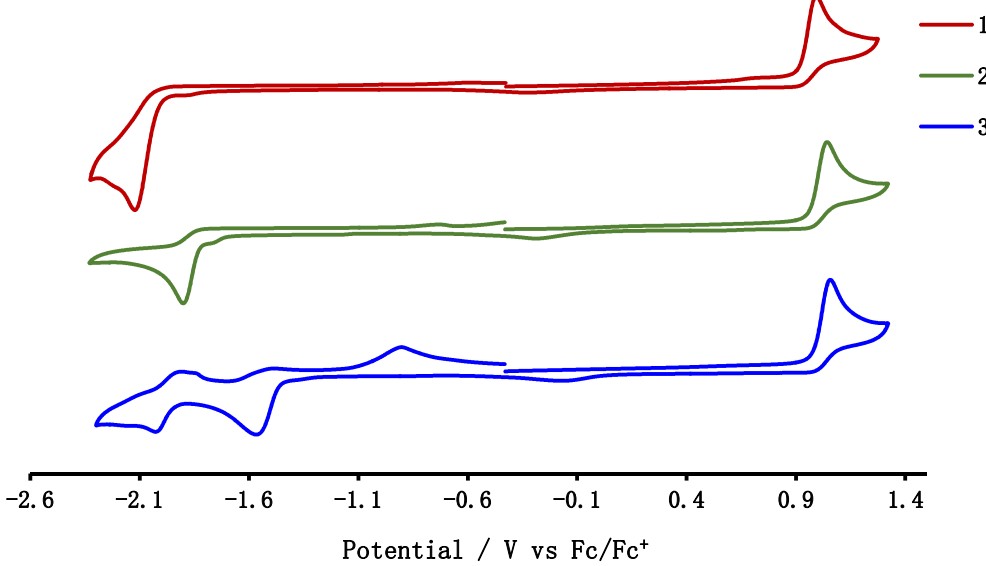

**Figure 2.** Cyclic voltammetry traces for complexes **1** to **3** in MeCN (1 mM of compounds **1**, **2**, and **3** in MeCN with 0.2 M Bu$_4$NPF$_6$ under nitrogen, recorded at 0.1 V·s$^{-1}$ at a glassy carbon disk electrode at room temperature).

## 2.3. Electronic Structure

The photophysical properties of the complexes were investigated and summarised data are provided in Table 2. UV/visible absorption spectra of **1** to **3** were recorded in acetonitrile solutions and are provided in Figure 3. The spectra exhibit strong absorptions below 300 nm ascribed to $\pi \rightarrow \pi^*$ N^N ligand-centred transitions with weaker bands at longer wavelengths assigned as singlet metal to ligand charge transfer ($^1$MLCT) transitions. These $^1$MLCT bands are red-shifted with respect to that of **1** ($\lambda_{max}$ = 335 nm) with the band for **2** appearing at 340 nm with that for **3** at 371 nm. This is consistent with the successive stabilisation of the pyrimidine- and pyrazine-localised LUMOs for **2** and **3**, respectively, relative to that of **1**.

**Table 2.** Summarised photophysical data for complexes **1** to **3** in acetonitrile.

| Complex | $\lambda_{abs}$/nm ($\varepsilon$/mol$^{-1}$ dm$^3$ cm$^{-1}$) | $\lambda^{max}_{em}$ R.T. $^a$ | 77 K $^b$ | $\tau$/ns $^a$ | $\Phi$/% $^{a,c}$ |
|---------|---------|---------|---------|---------|---------|
| 1 | 335 (3865), 296 (7690), 271 (11,530), 243 (18,150) | 540 $^d$ | 485 | 43 | 0.24 |
| 2 | 340 (3640), 268 (12,500), 247 (21,520) | 572 $^e$ | 505 | 55 | 0.25 |
| 3 | 371 (3640), 312 (8370), 285 (10,340), 231 (24,870) | 638 $^f$ | 545 | 92 | 0.38 |

$^a$ Aerated MeCN; $^b$ 4:1 EtOH/MeOH glass; $^c$ relative to [Ru(bpy)$_3$][PF$_6$]$_2$ in aerated MeCN ($\Phi$ = 1.8%), $^d$ $\lambda_{ex}$ = 350 nm; $^e$ $\lambda_{ex}$ = 365 nm; $^f$ $\lambda_{ex}$ = 390 nm.

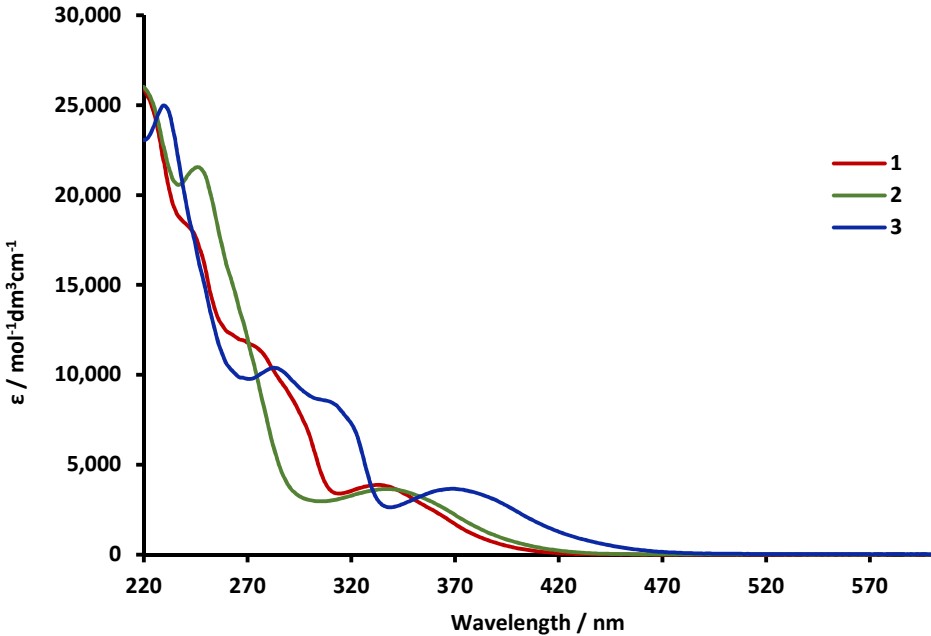

**Figure 3.** UV-Visible electronic absorption spectra for complexes **1** to **3** in acetonitrile at room temperature.

The complexes are luminescent in aerated acetonitrile solutions and exhibit broad featureless bands at significantly longer wavelengths than their $^1$MLCT absorption bands, indicative of emission from $^3$MLCT states (Figure 4a and Table 2). The emission spectra follow the same trend for the electronic absorption spectra with red-shifting of the $^3$MLCT bands in the order **1** (540 nm) < **2** (572 nm) < **3** (638 nm). This again reflects the progressive stabilisation of the pyrimidine and pyrazine centred LUMO in these complexes relative to that of **1** and mirrors the trend in $^3$MLCT/$^3$LL'CT state energies observed in iridium(III) complexes containing these ligands [36]. Emission spectra at 77 K revealed similar featureless $^3$MLCT-derived emission bands, which follow the same trend in energy as exhibited

at room temperature ($\lambda_{em}$ **1** < **2** < **3**), although with all emission maxima now shifted to higher energy owing to rigidochromic effects. The emission lifetimes for compounds **1**–**3** were recorded in aerated acetonitrile solutions and interestingly show reversed ordering to expectations based on energy-gap law considerations. In ruthenium(II) [40–44] and even osmium(II) [45,46] complexes, the presence of a 1,2,3-triazole-based chelate ligand can result in the quenching of luminescence and promotion of photochemical reactivity through increased accessibility of $^3$MC states from the photoexcited $^3$MLCT state. Indeed, the rhenium(I) complex [Re(btz)(CO)$_3$(Cl)] is non-emissive in fluid solution [47]. Further, the 4,4'-bi-1,2,3-triazolyl (btz) ligand has a significantly destabilised LUMO relative to the more commonly utilised 2,2'-bipyridyl chelate ligand by approximately 1 eV. The destabilisation of the LUMO in complexes of Ru(II) [44], Os(II) [46], and Ir(III) [48] through incorporation of btz has been shown to result in photochemical btz release in donor solvents. Pyridyltriazole-based ligands with intermediate destabilisation of the ligand-localised LUMO in comparison with btz have similarly been shown to undergo photochemical ligand release [15]. It is, therefore, entirely possible that in the complexes investigated here the $^3$MC state is thermally accessible from the photoexcited $^3$MLCT state. We therefore tentatively assign the inverted energy-gap law behaviour of complexes **1** to **3** as arising from increased separation of the $^3$MC and $^3$MLCT states as the ligand-based LUMO is progressively stabilised from **1** to **3**, thereby leading to elongated excited state lifetimes.

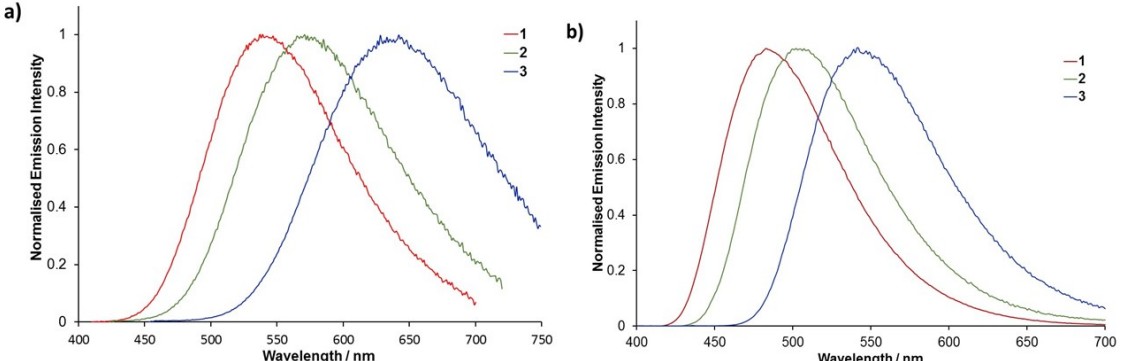

**Figure 4.** Normalised photoluminescence spectra for complexes **1** to **3** in aerated acetonitrile at room temperature (**a**) and in 4:1 EtOH/MeOH glass at 77 K (**b**).

In order to gain a more complete understanding of the photophysical and electrochemical properties imparted by the triazole-based ligands we turned to density functional theory (DFT) and time-dependant DFT calculations (TD-DFT analysis and main excitation energies for complexes **1**–**3** arereported in Supplementary Materials, Tables S1–S3). Optimised singlet ground state geometries for the three complexes **1**–**3** were calculated at the B3LYP level of theory using def2svp split valence polarization effective core potential and basis sets for the metal [49,50] and 6-311G++(d,p) basis sets for all other atoms (atomic coordinates for calculated ground states of complexes **1**–**3** are reported in the Supplementary Materials). Molecular orbital localisations (Figure 5) and energies (Table 3) were then determined in single-point calculations using the conductor-like polarizable continuum model (C-PCM) to simulate the solvent ($\varepsilon$ = 37.5 for acetonitrile). In common with the large number of [Re(N^N)(CO)$_3$Cl]-type complexes known in the literature, the HOMO of complexes **1**–**3** predictably has a significant metallic d-orbital character localised on the rhenium centre with additional contributions from CO $\pi^*$- and Cl p-orbitals. On the other hand, the LUMO orbitals are essentially localised on the chelate ligand. In agreement with the photophysical data, there is a significant stabilisation of the N^N ligand-based LUMO for **3** compared with complexes **1** and **2**. Thus, the HOMO–LUMO gap for **3** is about 0.46 eV smaller than for **1**, in excellent agreement with the electrochemical data (Table 1).

**Table 3.** Calculated HOMO and LUMO energies for complexes **1–3**.

| Complex | HOMO E/eV | LUMO E/eV | ΔE/eV |
|---------|-----------|-----------|-------|
| **1** | −6.35 | −2.32 | 4.02 |
| **2** | −6.46 | −2.57 | 3.89 |
| **3** | −6.42 | −2.85 | 3.57 |

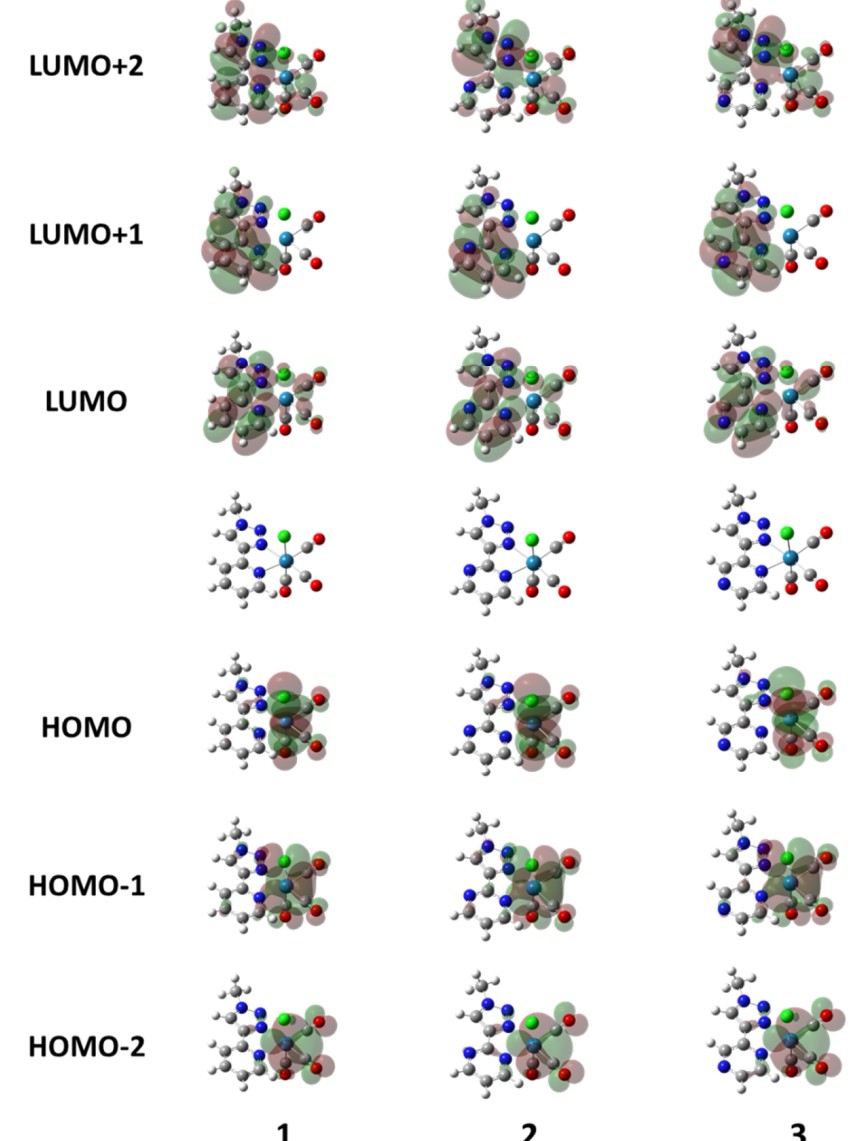

**Figure 5.** Optimised geometries and plots of HOMO (bottom) and LUMO (top) orbitals for complexes **1–3**.

The lowest triplet excited states for complexes **1** to **3** were optimised starting from their optimised singlet ground state geometries (atomic coordinates for calculated triplet states of complexes **1–3** are reported in the Supplementary Materials). The calculated spin densities are distributed over the both the Re(CO)$_3$Cl and N^N ligand fragments (Figure 6), as would be expected for a state of $^3$MLCT character. The energies of the $^3$MLCT states are progressively stabilised from **1** (2.35 eV) to **2** (2.30 eV) to **3** (2.11 eV) relative to their respective ground states, in agreement with experimental spectroscopic data. Analysis of the electrostatic potential (ESP) atomic charges for these $^3$MLCT states (compared with those of their corresponding ground states) shows that there is an accumulation of charge on

the NˆN ligand (by −0.48 to −0.63) with a corresponding depletion on the Re(CO)$_3$Cl fragment in agreement with the assignment of $^3$MLCT excited state character.

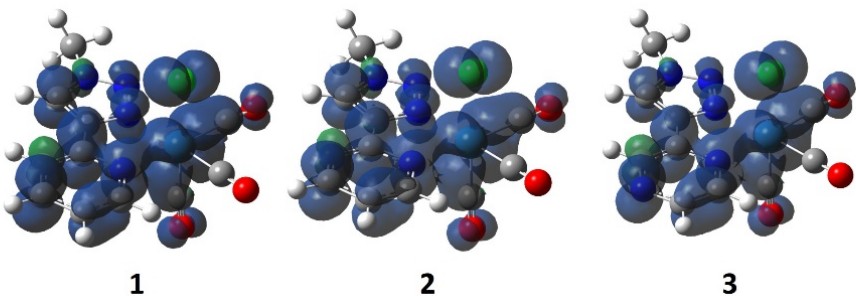

**Figure 6.** Plots of calculated spin densities for the lowest triplet excited state of complexes **1** to **3**.

## 2.4. CO$_2$ Electroreduction

A preliminary evaluation of the catalytic activity for **1** to **3** with respect to the electroreduction of CO$_2$ was undertaken using CV under conditions similar to those described earlier, but in CO$_2$-saturated solutions. A clear catalytic wave is observed in the case of **1** and **2** (Figure 7). In contrast, for reasons requiring further investigation, this wave was much less intense with **3** (Figure 7). By comparison of the shift and shape of the catalytic wave with those for similar rhenium carbonyl complexes recorded during CO$_2$ reduction [35,51], we can assume that the main reduction product of the systems here reported is CO. In the case of **3**, the first reduction feature occurs at more positive potential, shifting anodically in the presence of CO$_2$, possibly owing to a potential coordination of the pyrazinyl nitrogen by CO$_2$, thus making the complex more easily reduced [52,53]. In all cases (**1–3**), in the presence of CO$_2$, the catalytic wave is observed at a potential more negative (circa −2.3 V vs. Fc/Fc$^+$) than that corresponding to ligand reduction (Table 1). Consistently, addition of water has a stimulating effect on the catalytic wave by protonating the coordinated CO$_2$ molecules and facilitating their reduction to CO. As seen in Figure 7, in the case of complexes **1** and **2**, upon the addition of water the catalytic peak increases in intensity whereas for complex **3** this does not seem to affect the system. Instead, for complex **3**, the addition of water seems to anodically shift the reduction peak. This observation is tentatively interpreted as reflecting a displacement of the anionic chloride ligand by a neutral aqua ligand, thus making the complex more easily reduced [35].

In Table 1 the three complexes are compared in terms of redox potentials and $(i_c/i_p)^2$, where $i_c$ is the catalytic peak current and $i_p$ is the peak current associated with the catalyst in the absence of substrate [54]. The term $(i_c/i_p)^2$ is proportional to the turnover frequency (TOF), thus providing a useful benchmark of the relative TOF. Among the three complexes **1** resulted in a $(i_c/i_p)^2$ value of 15, in line with the results reported by Ching et al. under similar conditions [35] whereas **2** and **3** achieved values of 8 and 5 respectively.

To conclude, on the basis of the reported observations, complex **1** resulted in better catalytic performance than for the other two complexes. The results reported in this preliminary study thus indicate that the new complexes are inferior molecular designs for new electrocatalysts for CO$_2$ reduction; however, future controlled potential electrolysis experiments will be required to fully verify and consolidate these preliminary data.

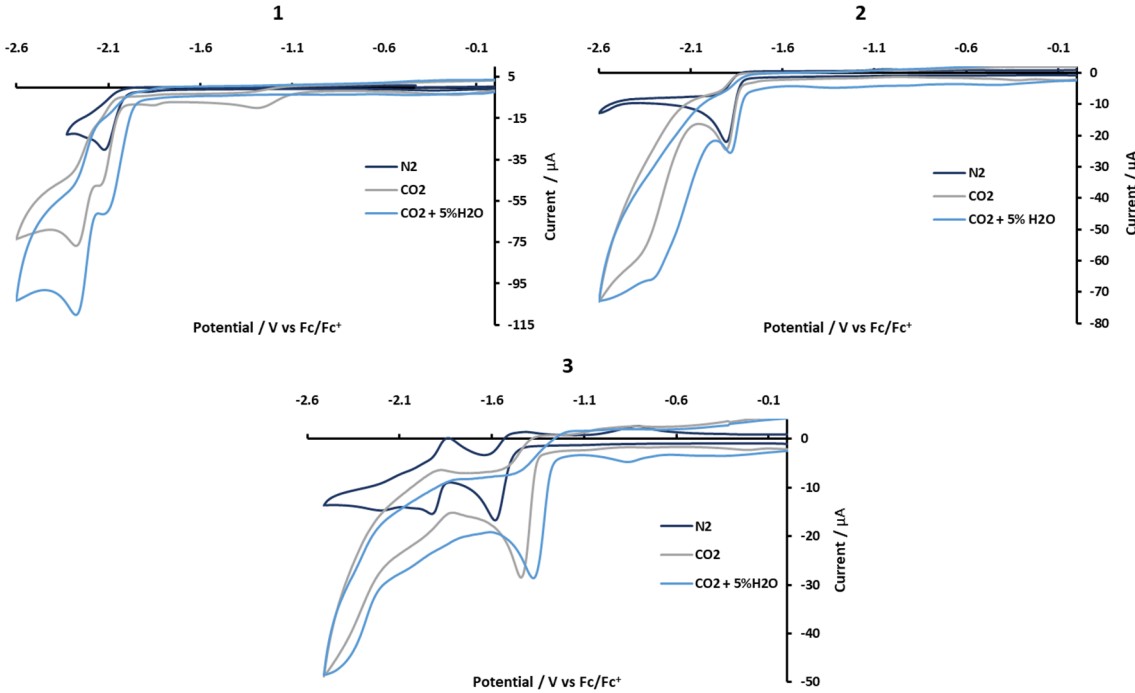

**Figure 7.** Cyclic voltammetry traces for complexes **1** to **3** in acetonitrile with 0.1 M Bu$_4$NPF$_6$ under nitrogen or CO$_2$ and with or without water, recorded at 0.1 V·s$^{-1}$ at a glassy carbon disk electrode at room temperature.

## 3. Experimental Section

Chemicals were purchased from Sigma-Aldrich (Dorset, UK) and Acros Organics (Fisher Scientific UK, Loughborough, UK). $^1$H NMR and $^{13}$C NMR spectra were recorded on a Bruker Avance 400 MHz instrument (Bruker, Rheinstetten, Germany). Mass spectrometry data were collected on a Bruker Micro Q-TOF instrument (Bruker, Hamburg, Germany). UV/visible absorption spectra were recorded on a Varian Cary 300 spectrophotometer (Varian Inc., Crawley, UK) and corrected emission spectra were recorded on a Horiba Fluoromax-4 spectrofluorometer (Horiba, Kyoto, Japan). Luminescent lifetime measurements were carried out using an Edinburgh Instruments Mini-Tau spectrometer (Edinburgh Instruments, Livingston, UK). Complex **1** was prepared following the previously reported procedure [16].

### 3.1. Synthesis of [Re(pymtz)(CO)$_3$Cl (**2**)

2-(1-Benzyl-1*H*-1,2,3-triazol-4-yl)-pyrimidine (151 mg, 0.64 mmol) and Re(CO)$_5$Cl (232 mg, 0.64 mmol) were suspended in toluene (80 mL) and refluxed at 110 °C overnight before cooling to room temperature. Excess hexane was added to the dark yellow-coloured solution and the resulting precipitate was collected by vacuum filtration. The solids were subsequently re-dissolved in dichloromethane and filtered through a celite plug. Evaporation of the filtrate afforded a pale-yellow coloured solid. Yield = 0.235 g, 68%.

$^1$H NMR (*d*$_3$-MeCN, 400 MHz): 5.72 (d, *J* = 15.1 Hz, 1H, *CH$_2$*), 5.77 (d, *J* = 15.1 Hz, 1H, *CH$_2$*), 7.39–7.48 (m, 5H, *Bn*), 7.52 (t, *J* = 5.3 Hz, 1H, *pym*), 8.70 (s, 1H, *tz*), 9.00 (dd, *J* = 2.1, 5.2 Hz, 1H, *pym*), 9.12 (dd, *J* = 2.1, 5.7 Hz, 1H, *pym*). $^{13}$C NMR (*d*$_3$-MeCN, 101 MHz): 56.50, 122.73, 128.13, 129.56, 130.11, 130.13, 134.63, 148.11, 160.17, 160.92, 160.99. HRMS (ESI): *m/z* calc. for [C$_{16}$H$_{11}$N$_5$O$_3$Re]$^+$ calc = 508.0419, found = 508.0418 (M − Cl)$^+$; [C$_{18}$H$_{14}$N$_6$O$_3$Re]$^+$ calc = 549.0685, found = 549.0686 (M − Cl + MeCN)$^+$.

## 3.2. Synthesis of [Re(pyztz)(CO)₃Cl] (**3**)

2-(1-Benzyl-1*H*-1,2,3-triazol-4-yl)-pyrazine (100 mg, 0.42 mmol) and Re(CO)₅Cl (153 mg, 0.42 mmol) were added to toluene (25 mL) and heated to 100 °C, giving a clear orange-coloured solution, for 24 h. The solution was allowed to cool to room temperature and hexane (20 mL) was added to ensure complete precipitation of the product. The yellow solids were collected by filtration and washed with ether. Yield = 199 mg, 87% $^1$H NMR ($d_3$-MeCN, 400 MHz): 5.75 (d, $J$ = 14.6 Hz, 1H, $CH_2$), 5.79 (d, $J$ = 14.6 Hz, 1H $CH_2$), 7.42–7.50 (m, 5H, *Bn*), 8.66 (s, 1H, *tz*), 8.70 (d, $J$ = 3.1 Hz, 1H, *pyz*), 8.92 (dd, $J$ = 1.3, 3.1 Hz, 1H *pyz*), 9.25 (d, $J$ = 1.1 Hz, 1H, *pyz*). $^{13}$C NMR ($d_3$-MeCN, 101 MHz): 56.55, 126.41, 129.77, 130.22, 130.23, 134.45, 145.02, 145.26, 147.21, 147.80. HRMS (ESI): *m/z* calc. for $[C_{16}H_{11}N_5O_3Re]^+$ calc = 508.0419, found = 508.0413 $(M - Cl)^+$; $[C_{18}H_{14}N_6O_3Re]^+$ calc = 549.0685, found = 549.0666 $(M - Cl + MeCN)^+$.

## 3.3. Electrochemistry

Electrochemical measurements were performed using a PalmSens EmStat electrochemical workstation (PalmSens, Houten, The Netherlands). A single-compartment cell was used for all cyclic voltammetry experiments with a 1.0 mm diameter glassy carbon disk working electrode, a platinum wire counter electrode, a Ag/AgCl reference electrode and 0.2 M tetrabutylammonium hexafluorophosphate (TBAPF₆) as the supporting electrolyte in acetonitrile. The ferrocene/ferrocenium redox couple was used as an internal standard. TBAPF₆ was recrystallized from ethanol and oven-dried prior to use. Electrochemical cells were shielded from light during experiments. All solutions were continuously purged with nitrogen before the cyclic voltammetry experiments, with measurements recorded under an atmosphere of N₂. Solutions of rhenium complexes were prepared in concentrations of 1.0 mM for cyclic voltammetry experiments. Solutions were purged with CO₂ for 30 min prior to catalytic experiments.

## 3.4. Computational Methods

DFT calculations were performed using Gaussian 09 software. Geometry optimizations were performed at B3LYP/6-311G++(d,p) level of theory in the presence of polarizable continuum acetonitrile solvent implemented using the C-PCM solvation model in Gaussian 09. A split valence polarization effective core potential basis (def2svp) [55] was used for the Re atom in all the above-mentioned calculations. To further evaluate the vertical transitions, ωB97XD/def2tzvp [56] calculations were performed on the B3LYP/def2svp structures. To reduce computational cost, the benzyl groups were replaced by methyl groups, as the benzyl substituent does not affect the spectroscopic properties of such complexes.

## 4. Conclusions

A new series of rhenium(I) triazole tricarbonyl complexes were prepared and studied for their photophysical behaviour and their ability to catalyse the electroreduction of CO₂. We specifically investigated the influence of the two isomeric pyrimidine/pyrazine–triazolyl ligands, with respect to the previously known pyridyltriazole ligand, on the electronic and spectroscopic properties of the corresponding rhenium complexes. The inclusion of an additional nitrogen atom to the six-membered heterocyclic ring in **2** and **3** leads to a stabilisation of the LUMO with respect to that of **1**. This results in a red-shift of the optical absorption bands and a lowering in energy of photoluminescence. This also interestingly leads to extended excited state lifetime and increased quantum yield contrary to the energy gap law. As the triazole-based ligands have previously been shown to encourage luminescence quenching and even photochemical reactivity through increased accessibility of $^3$MC states, this behaviour was ascribed to reduced accessibility of $^3$MC states upon $^3$MLCT state stabilisation as the six-membered heterocycle becomes more electron withdrawing. Complexes **1** and **2** showed a modest electrocatalytic activity in the presence of CO₂, whereas complex **3** did not show an appreciable

catalytic wave under the same experimental conditions. For complexes **1** and **2**, addition of water as a proton source enhanced the intensity of the catalytic reduction wave. This work thus offers insights into the spectroscopic and catalytic structure–property relationships and adds new compounds to the growing family of rhenium triazole complexes.

**Supplementary Materials:** The following are available online at http://www.mdpi.com/2304-6740/8/3/22/s1, Figure S1: $^1$H NMR (400 MHz, $d_3$-MeCN) spectrum of complex **2**; Figure S2: $^{13}$C NMR (101 MHz, $d_3$-MeCN) spectrum of complex **2**; Figure S3: $^1$H NMR (400 MHz, $d_3$-MeCN) spectrum of complex **3**; Figure S4: $^{13}$C NMR (101 MHz, $d_3$-MeCN) spectrum of complex **3**; Table S1: Main excitation energies and oscillator strengths for complex **1** (orbital 79 = HOMO and orbital 80 = LUMO); Table S2: Main excitation energies and oscillator strengths for complex **2** (orbital 79 = HOMO and orbital 80 = LUMO); Table S3: Main excitation energies and oscillator strengths for complex **3** (orbital 79 = HOMO and orbital 80 = LUMO).

**Author Contributions:** Conceptualization, P.I.P.E., P.A.S., A.S.; validation, P.A.S., A.S.; formal analysis, A.S.; investigation, P.A.S., A.S., A.C., L.C., S.A.E.O.; writing—original draft preparation, P.I.P.E., P.A.S., A.S.; writing—review and editing, P.I.P.E., P.A.S., A.S.; supervision, P.I.P.E., P.A.S. All authors have read and agreed to the published version of the manuscript.

**Funding:** This research was funded by Qatar National Research Fund, a member of Qatar Foundation, grant number NPRP9-174-2-092.

**Acknowledgments:** The authors wish to thank the University of Huddersfield and QNRF (Qatar National Research Fund, a member of Qatar Foundation) for funding this research. We are grateful to the Research Computing Center in Texas A& M University at Qatar, where the calculations were conducted. We thank Mohamed E. Madjet for the support in improving the computational section.

**Conflicts of Interest:** The authors declare no conflict of interest.

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
