# Peer review of "Photophysical and Electrocatalytic Properties of Rhenium(I) Triazole-Based Complexes"

_inorganics, doi:10.3390/inorganics8030022_

Round 1

Reviewer 1 Report

The authors describe the synthesis and the characterization of three mononuclear containing a rhenium complexes containing a triazole-based ligand. Even if only two of the three complexes presented here are new and even if the performances of these complexes as electrochemical catalysts for the reduction of CO2 are quite low, the spectroscopic and the electrochemical characterizations are well done and in agreement with computational data. For these reasons this paper is suitable for the publication with very minor revisions.

In particular:

  1. Page 2 line 90: the Authors attribute the second reduction peak of complex 3 to a further reduction of the pyrazine moiety. However in some other rhenium complexes the second reduction peak is due to the reduction of metal center from Re(I) to Re(0). Did the authors consider this possibility? Could the Authors add some comments regarding this point?
  2. Page 4 line 120: please change l 1>2>3 with l 1<2<3 since the emission maximum of complex 1 is lower than that of complex 2 and complex 3, not vice versa.

Author Response

Many thanks for your complementary comments on our manuscript and for the extremely helpful suggestions for its improvement. We have addressed the comments, we outline our point by point responses below to your request and have uploaded an updated manuscript highlighted to show changes made.

  • Page 2 line 90: the Authors attribute the second reduction peak of complex 3 to a further reduction of the pyrazine moiety. However in some other rhenium complexes the second reduction peak is due to the reduction of metal center from Re(I) to Re(0). Did the authors consider this possibility? Could the Authors add some comments regarding this point?
  • Very valuable comment, in fact we believe our assumption was most probably wrong. The CV of the isolated ligand showed only 1 reduction wave. After comparison with similar reports in literature,  we possibly assign that reduction wave to Re(I/0). 

  • Page 4 line 120: please change l 1>2>3 with l 1<2<3 since the emission maximum of complex 1 is lower than that of complex 2 and complex 3, not vice versa.
  • The requested correction has been made.

Reviewer 2 Report

The manuscript describes synthesis of  new rhenium complexes with diimine ligands, and a study of their photo- and electrochemical properties. The aim has been to design coordination complexes that could perform well in catalytic reduction of CO2. The topic is important because of the growing concern on the global climate change, where every small step to solve the problem is valuable. Several types of transition metal complexes have been proven to serve as active catalysts in CO2 reduction, but still there is room for new improvements, which can be obtained via ligand design.

For the most part, the manuscript is well written and easy to read, but I have a few major comments, which the authors should consider before I can recommend accepting the paper in Inorganics.

1) It would help a lot to follow the story, if the authors would include subtitles in the Results section.

2) I am a bit concerned on the conclusions made on the emission spectra, where the authors claim the observed trends arise from increased separation of the two different triplet states. Do they have any proof on the nature of the triplet states. I would suggest to clarify this point computationally.

3) I am also concerned on the quality of the DFT calculations. First of all, it is well known that B3LYP (especially when combined with the Los Alamos basis set for the metal) does not perform very well for transition metal complexes, and the reliability is especially poor when comparing MO energies. Did the authors do test calculations or where did they base their selection? The selection of the method should be at least justified in the text.

4) The appearance of the FMOs is not very clear. First they could reduce the isovalue, when making the pictures, to help to see where the orbitals are really concentrating. If they could perform a fragment analysis (by calculating the % contribution of selected atomic orbitals in the MO), they could give more specific explanation on the reason of the reduced energy gap.

5) Figure 6 is completely unnecessary, since the authors do not make any additional conclusions that cannot be seen from Table 3.

Author Response

Thanks for your complementary comments on our manuscript and for the extremely helpful suggestions for its improvement. We have addressed most of the comments, we outlined our point by point responses below to your request and have uploaded an updated manuscript, where changes made are highlighted.

  • It would help a lot to follow the story, if the authors would include subtitles in the Results section.
  • Subsections have been added as requested.

  • I am a bit concerned on the conclusions made on the emission spectra, where the authors claim the observed trends arise from increased separation of the two different triplet states. Do they have any proof on the nature of the triplet states. I would suggest to clarify this point computationally.
  • Several works in the literature for metal complexes bearing triazole ligands have reported that the 3MC is thermally accessible from the photoexcited 3MLCT state. We have added to the existing discussion on this point with further references. The lowest triplet excited states for complexes 1 to 3 have additionally been optimised in additional work in our revisions to corroborate the conclusions and the corresponding calculated spin densities are distributed over the both the Re(CO)3Cl and N^N ligand fragments as would be expected for a state of 3MLCT character. Changes in partial charges on these fragments are consistent with 3MLCT character of the T1 The energies of the 3MLCT states have been added to the manuscript and they also confirmed a progressive stabilisation along the series, relative to their respective ground states in agreement with experimental spectroscopic data. Further discussion on this point has been added to the main manuscript. Recent reports have detailed the previously unappreciated complexity of the 3MC region of triplet excited state potential energy surfaces of metal complexes. We feel that the calculation of all possible 3MC states in these complexes will be challenging task in itself and also beyond the immediate scope of the current article. Based on the results presented and the literature precedents we feel that the inverted energy gap law behaviour is justifiably explained through the 3MLCT state of the pyridine-containing complex undergoing thermal quenching of emission via 3MC states to a greater degree than the pyrazine-containing complex. However, we have rephrased the text to state that this is a “tentative” assignment.

  • I am also concerned on the quality of the DFT calculations. First of all, it is well known that B3LYP (especially when combined with the Los Alamos basis set for the metal) does not perform very well for transition metal complexes, and the reliability is especially poor when comparing MO energies. Did the authors do test calculations or where did they base their selection? The selection of the method should be at least justified in the text.
  • Thanks for this constructive point, we therefore changed the basis set based on those used for similar studies in the literature. B3LYP/6-311G++(d,p) level of theory was used for C, N, O,Cl and H, whereas def2svp was used for the Re atom. To further evaluate the vertical transitions, ωB97XD/def2tzvp calculations were performed on the B3LYP/def2svp structures. New calculations and methods are reported in the manuscript.

4) The appearance of the FMOs is not very clear. First they could reduce the isovalue, when making the pictures, to help to see where the orbitals are really concentrating. If they could perform a fragment analysis (by calculating the % contribution of selected atomic orbitals in the MO), they could give more specific explanation on the reason of the reduced energy gap.

  • The appearance of the FMOs has been modified for a greater clarity and now clearly demonstrate the localisation of occupied FMOs over the Re(CO)3Cl fragment with the lowest virtual FMOs localised over the N^N. We have added some further discussion drawing upon our recent report on the iridium(III) complexes of the same N^N ligands which shows the same trend in emission and excited state energies based on the electron withdrawing character of the 6-membered heterocycle. Analysis of the EDS atomic charges for these 3MLCT states (compared to those of their corresponding ground states) has been reported in the text to show that is the accumulation of charge on the N^N ligand with a corresponding depletion on the Re(CO)3Cl fragment, in agreement with the assignment of 3MLCT excited state character.

  • Figure 6 is completely unnecessary, since the authors do not make any additional conclusions that cannot be seen from Table 3.
  • Figure 6 has been removed accordingly and replaced with a plot of spin densities based on the referee’s earlier comments.

Round 2

Reviewer 2 Report

All comments have been addressed and I can recommend publication in the present form